# A gut bacterial amyloid promotes α-synuclein aggregation and motor impairment in mice

Timothy R Sampson[1‡]*, Collin Challis[1†], Neha Jain[2†§], Anastasiya Moiseyenko[1], Mark S Ladinsky[1], Gauri G Shastri[1], Taren Thron[1], Brittany D Needham[1], Istvan Horvath[3], Justine W Debelius[4#], Stefan Janssen[4¶], Rob Knight[4,5], Pernilla Wittung-Stafshede[3], Viviana Gradinaru[1], Matthew Chapman[2], Sarkis K Mazmanian[1]*

[1]Division of Biology & Biological Engineering, California Institute of Technology, Pasadena, United States; [2]Department of Molecular, Cellular, and Developmental Biology, University of Michigan, Ann Arbor, United States; [3]Department of Biology and Biological Engineering, Chalmers University of Technology, Gothenburg, Sweden; [4]Department of Pediatrics, University of California, San Diego, San Diego, United States; [5]Department of Computer Science and Engineering, University of California, San Diego, San Diego, United States

*For correspondence:
trsamps@emory.edu (TRS);
sarkis@caltech.edu (SKM)

[†]These authors contributed equally to this work

Present address: [‡]Department of Physiology, Emory University School of Medicine, Atlanta, United States; [§]Department of Bioscience and Bioengineering, Indian Institute of Technology, Jodhpur, India; [#]Department of Microbiology, Tumor and Cell Biology, Karolinska Institutet, Stockholm, Sweden; [¶]Algorithmic Bioinformatics, Justus-Liebig-University Giessen, Giessen, Germany

**Abstract** Amyloids are a class of protein with unique self-aggregation properties, and their aberrant accumulation can lead to cellular dysfunctions associated with neurodegenerative diseases. While genetic and environmental factors can influence amyloid formation, molecular triggers and/or facilitators are not well defined. Growing evidence suggests that non-identical amyloid proteins may accelerate reciprocal amyloid aggregation in a prion-like fashion. While humans encode ~30 amyloidogenic proteins, the gut microbiome also produces functional amyloids. For example, curli are cell surface amyloid proteins abundantly expressed by certain gut bacteria. In mice overexpressing the human amyloid α-synuclein (αSyn), we reveal that colonization with curli-producing *Escherichia coli* promotes αSyn pathology in the gut and the brain. Curli expression is required for *E. coli* to exacerbate αSyn-induced behavioral deficits, including intestinal and motor impairments. Purified curli subunits accelerate αSyn aggregation in biochemical assays, while oral treatment of mice with a gut-restricted amyloid inhibitor prevents curli-mediated acceleration of pathology and behavioral abnormalities. We propose that exposure to microbial amyloids in the gastrointestinal tract can accelerate αSyn aggregation and disease in the gut and the brain.

## Introduction

The accumulation and aggregation of amyloid proteins occurs during many neurodegenerative diseases. Synucleinopathies are one family of amyloid disease, which includes Parkinson's disease (PD), Lewy Body disease (LBD), and Multiple System Atrophy (MSA). Central to their pathogenesis is the accumulation of the neuronal protein α-Synuclein (αSyn) into insoluble amyloid aggregations, which ultimately leads to inflammation and neuronal dysfunction (*Jucker and Walker, 2013*; *Brettschneider et al., 2015*). αSyn aggregation can contribute to the death of dopaminergic neurons in specific brain regions, resulting in motor symptoms (*Poewe et al., 2017*). Clinical and epidemiological data suggest that accumulation of αSyn aggregates may first occur at peripheral sites, such as the olfactory epithelium or gastrointestinal (GI) tract, before spreading to the brain (*Braak et al., 2003*). Individuals diagnosed with various synucleinopathies often display constipation

and other GI dysfunctions years prior to the onset of movement dysfunction (*Verbaan et al., 2007*; *Colosimo, 2011*; *Engen et al., 2017*; *Mertsalmi et al., 2017*; *Sakakibara et al., 2019*). Experimental evidence exists for a prion-like spread of αSyn aggregates (*Woerman et al., 2015*), including propagation from the gut to the brain via the vagus nerve and/or spinal cord in rodent models (*Holmqvist et al., 2014*; *Uemura et al., 2018*; *Van Den Berge et al., 2019*). In humans, recent epidemiological studies suggest an association between truncal vagotomy and appendectomy with a decreased risk of PD (*Svensson et al., 2015*; *Liu et al., 2017*; *Killinger et al., 2018*), and in increased risk of comorbidity with inflammatory bowel disease (IBD) (*Hui et al., 2018*; *Peter et al., 2018*). While a role for protein aggregation and/or inflammation in the gut represents an emerging area of research in synucleinopathies, the GI tract has been implicated in other neurological disorders such as autism spectrum disorder, depression, anxiety and Alzheimer's disease (*Vuong et al., 2017*).

The gut is colonized with a complex microbiome that impacts development and function of the immune, metabolic and nervous systems (*Fung et al., 2017*). Enterobacteriaceae, highly prevalent within the gut of humans, can produce functional amyloid proteins termed curli (*Tursi and Tükel, 2018*). Curli fibers are formed by bacterial secretion of an unfolded amyloid, CsgA, that subsequently aggregates extracellularly to form biofilms, mediate adhesion to epithelial cells, and are involved in bacteriophage defense (*Tursi and Tükel, 2018*; *Vidakovic et al., 2018*). Exposure to curli not only modulates host inflammatory responses within the intestinal tract and periphery (*Gallo et al., 2015*; *Chen et al., 2016*; *Tursi and Tükel, 2018*), but oral administration of curli-producing bacteria can also increase production and aggregation of the amyloid protein αSyn in aged rats and nematodes (*Chen et al., 2016*). Biochemical studies demonstrate that native, bacterial chaperones of curli are capable of transiently interacting with αSyn and modulating its aggregation (*Chorell et al., 2015*; *Evans et al., 2015*). Interestingly, diverse human amyloid proteins including αSyn, amyloid beta (Aβ), cellular prion protein (PrP$^C$), and Tau can accelerate the amyloidogenesis of heterologous mammalian amyloid proteins (*Clinton et al., 2010*; *Brettschneider et al., 2015*; *Katorcha et al., 2017*). Lesions containing mixed human amyloids have been observed in neurodegenerative brains (*Rahimi and Kovacs, 2014*; *Spires-Jones et al., 2017*), implicating interactions between different amyloidogenic proteins in resulting pathology. Accordingly, we wondered whether a bacterial amyloid protein can contribute to heterologous aggregation of mammalian αSyn in the gut and the brain, leading to synucleinopathy-related behaviors.

Herein, we reveal that mono-colonization of αSyn-overexpressing mice with curli-producing *Escherichia coli* exacerbates motor impairment and GI dysfunction, and promotes αSyn aggregation and inflammation in the gut and brain. Enrichment of curli-producing *E. coli* to mice harboring a healthy human microbiome is sufficient to aggravate αSyn-dependent pathophysiology. The purified amyloidogenic subunit of curli fibers (CsgA) is sufficient to accelerate αSyn aggregation during in vitro biochemical assays and pathophysiology in mice following intra-intestinal administration, while variants of CsgA that are unable to form amyloids have no effect on αSyn aggregation. Oral treatment of mice with a gut-restricted amyloid inhibitor reduces *csgA* expression in the gut, limits αSyn aggregation in the brain, and alleviates GI and motor deficits in mice that overexpress αSyn. These data provide novel insights into a trans-kingdom interaction between the gut microbiome and mammalian amyloids, and suggest the possibility that carriage of particular bacterial taxa may be a factor that can exacerbate neurologic disease.

## Results and discussion

### Mono-colonization with curli-producing gut bacteria enhances αSyn pathophysiology

We previously identified that depletion of the microbiome reduces pathophysiology in Thy1-αSyn mice (alpha-synuclein overexpressing; ASO mice) (*Sampson et al., 2016*), which overexpress wildtype human αSyn. Due to accumulation and aggregation of neuronal αSyn, mice display increased neuroinflammation, GI dysfunction, and progressive motor abnormalities (*Rockenstein et al., 2002*; *Chesselet et al., 2012*; *Wang et al., 2012*) that are relevant in the study of synucleinopathies. Prior findings in germ-free Thy1-αSyn mice suggest that an unidentified member(s) of the gut microbiome may be pathogenic in this mouse model (*Sampson et al., 2016*). Interestingly, increased colonization

and mucosal association with Enterobacteriaceae, such as *E. coli*, have been reported in individuals with PD compared to healthy controls (*Forsyth et al., 2011*), as well as a positive association of Enterobacteriaceae abundance with disease severity (*Scheperjans et al., 2015*; *Li et al., 2017*).

To establish whether *E. coli* promotes αSyn-dependent motor dysfunction, we mono-associated germ-free (GF) wild-type and ASO mice with the curli-producing *E. coli* strain MC4100, or *Bacteroides fragilis* strain NCTC9343 and segmented filamentous bacteria (SFB), which do not produce curli. *E. coli* exacerbated the αSyn-dependent motor defects in ASO animals across a battery of tests, compared to the other taxa (*Figure 1A–F* and *Figure 1—figure supplement 1A–E*). To determine the contribution of curli amyloids, we compared mice mono-colonized with wild-type *E. coli* (WT) to those mono-colonized with an isogenic mutant lacking genes encoding the curli biosynthesis machinery (Δ*csgBAC*) (*Wang and Chapman, 2008*). Evaluation of coordinated motor function revealed that colonization with the curli-deficient strain did not elicit robust motor impairment (*Figure 1A–F*). We did not observe curli-dependent alterations to colonization levels or mucosal association, despite detecting *csgA* expression (*Figure 1—figure supplement 1F–I*). In addition, the curli-deficient mutant did not display alterations to lipopolysaccharide potency or structure (*Figure 1—figure supplement 1J,K*). Thus, curli-producing bacteria are capable of enhancing motor deficits in ASO mice.

ASO mice colonized with curli-producing bacteria displayed increased αSyn fibril reactivity and detergent-insoluble αSyn in the midbrain compared to mice with the Δ*csgBAC* strain, despite similar transgene production (*Figure 1—figure supplement 2A–D,F,G*). Histological and western blot analysis revealed increased phospho-serine129 αSyn (pS129-αSyn) deposition in the substantia nigra (SN) of ASO mice, indicative of pathological αSyn aggregation, (*Figure 1G,H*, and *Figure 1—figure supplement 2E,H,J*). Further, mice colonized with WT *E. coli* show increased proteinase K-resistant αSyn inclusions in the midbrain (*Figure 1I* and *Figure 1—figure supplement 2I*), with little alteration to pS129-αSyn in the frontal cortex (*Figure 1—figure supplement 2K,L*). Additionally, we observed increased pS129-αSyn in the proximal large intestine, and elevated αSyn fibril reactivity in the duodenum and proximal large intestine of ASO mice mono-colonized with curli-producing *E. coli* (*Figure 1—figure supplement 2M–P*). Previous research has shown that synuclein pathology, neuroinflammation, and motor defects occur at early ages in the ASO mouse model without concomitant loss of striatal dopamine, tyrosine hydroxylase positive (TH+) neurons in the midbrain, or loss of neurons in the myenteric plexus (*Fleming et al., 2004*; *Lam et al., 2011*; *Chesselet et al., 2012*; *Wang et al., 2012*). Consistent with these prior observations, we do not observe decreases in total striatal dopamine, midbrain *TH* expression, TH+ neurons, nor myenteric PGP9.5+ neurons, under the colonization conditions and ages assessed, suggesting pathology and motor deficits are independent of dopamine or neuron loss (*Figure 1*, figure supplement S2Q-S).

Mono-colonization of ASO mice with curli-producing *E. coli* resulted in increased expression of the proinflammatory cytokines interleukin 6 (IL-6) and tumor necrosis factor alpha (TNFα) in brain-derived CD11b+ cells, and increased cytokine production in the midbrain and striatum (*Figure 1—figure supplement 3A–C*). In addition, Iba+ microglia morphologies indicated reduced activation in ASO mice colonized with the curli-deficient strain (*Figure 1—figure supplement 3D–G*). Multiplexed ELISA analysis revealed increased cytokine and chemokine production in colonic tissue of mice mono-colonized with curli-producing *E. coli*, irrespective of genotype (*Figure 1—figure supplement 3H,J,K*), and no noteworthy changes to serum cytokines of ASO mice based on colonization status (*Figure 1—figure supplement 3I*). These data reveal that a bacterial amyloid from the gut microbiome can exacerbate pathology and inflammation, in both the gut and brain, in mice that overexpress αSyn.

## Curli biosynthesis within a complex microbiome contributes to motor and GI deficits

To explore host-microbiome interactions in a more natural context, we tested whether introduction of curli-producing bacteria to a healthy human microbiota is sufficient to enhance αSyn-dependent pathophysiology. GF ASO mice were transplanted with fecal microbiota from a human donor predicted to contain low levels of *csgA*, as indicated by PICRUSt analysis following 16S rRNA sequencing (*Figure 2—figure supplement 1A*). This single microbiome was supplemented with either WT *E. coli* or the Δ*csgBAC* strain. Both strains reached similar abundances in the feces (*Figure 2—figure supplement 1B,C*), while *csgA* expression and amyloid production appeared only in mice colonized

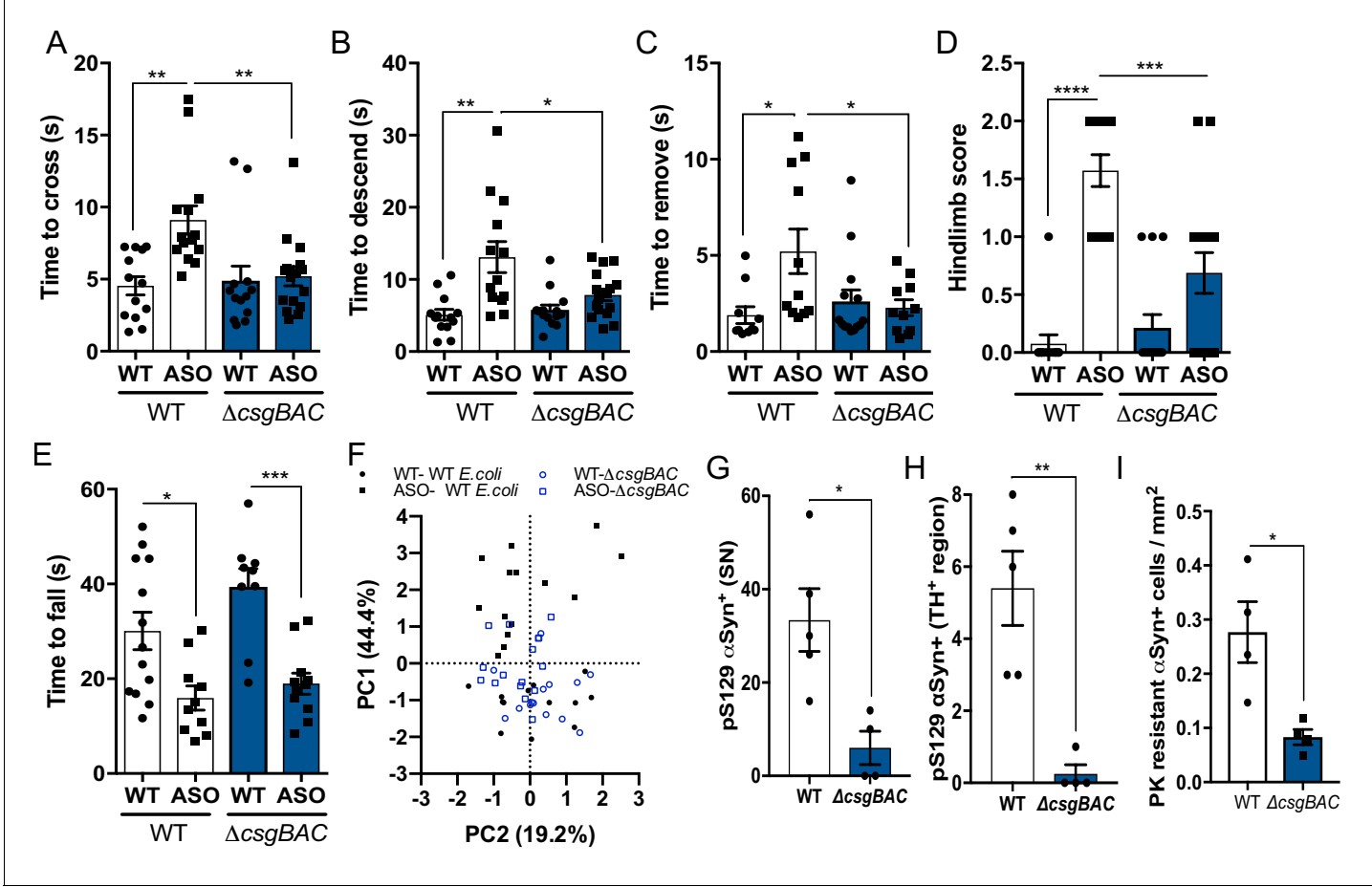

**Figure 1.** Mono-colonization with curli-producing gut bacteria enhances αSyn pathophysiology germ-free wild-type BDF1 (WT) and Thy1-αSyn (ASO) mice were mono-colonized with wild-type *E. coli* MC4100 (WT) or an isogenic curli-deficient strain (Δ*csgBAC*) at 5–6 weeks of age. Motor function was assessed at 12–13 weeks of age by quantifying (**A**), beam traversal time, (**B**), pole descent time, (**C**), nasal adhesive removal time, (**D**), hindlimb clasping score, (**E**), wirehang time. (**F**), Principal component analysis of compiled motor scores from tests in (**A–E**). Quantification of (**G, H**) pS129αSyn and (**I**) proteinase K–resistant αSyn by immunofluorescence microscopy in the substantia nigra and midbrain. n = 13–16 (**A–F**), n = 4.5 (**G–I**). Points represent individuals, bars represent the mean and standard error. Data analyzed by one-way ANOVA with Tukey post-hoc test (**A–E**), and two-tailed *t*-test for (**G–I**). *p≤0.05; **p≤0.01; ***p≤0.001; ****p≤0.0001. Motor data are compiled from three independent cohorts.

The online version of this article includes the following source data and figure supplement(s) for figure 1:

**Source data 1.** Source data and statistical analysis.

**Figure supplement 1.** *E. coli* alters motor deficits, and CsgA does not influence colonization, inflammatory capacity, or dopamine production Germ-free (GF) wild-type (WT) or Thy1- αSyn (ASO) mice were mono-colonized with either *Bacteroides fragilis* (Bfrag), segmented filamentous bacteria (SFB), or *Escherichia coli* (Ecoli).

**Figure supplement 1—source data 1.** Source data and statistical analysis.

**Figure supplement 2.** Mono-colonization with curli-sufficient bacteria induce increased αSyn-dependent pathology Germ-free (GF) wild-type (WT) or Thy1-αSyn (ASO) animals were mono-colonized with curli-sufficient *E. coli* (WT) or curli-deficient *E. coli* (Δ*csgBAC*).

**Figure supplement 2—source data 1.** Source data and statistical analysis.

**Figure supplement 3.** Mono-colonization with curli-sufficient bacteria induces altered inflammatory responses in the brain and intestine.

**Figure supplement 3—source data 1.** Source data and statistical analysis.

with WT bacteria (*Figure 2—figure supplement 1C–E*). Mice harboring a microbiota containing WT *E. coli* displayed significantly impaired motor and GI performance compared to animals with a complex microbiota plus the curli-deficient strain (*Figure 2A–G*). Moreover, enrichment of curli-producing bacteria resulted in increased αSyn fibril reactivity in the midbrain and elevated pS129-αSyn deposition in the SN, without changes to the number of TH[+] neurons (*Figure 2H* and *Figure 2—figure supplement 1F–J*). Morphometric analysis of microglia demonstrated concomitant changes

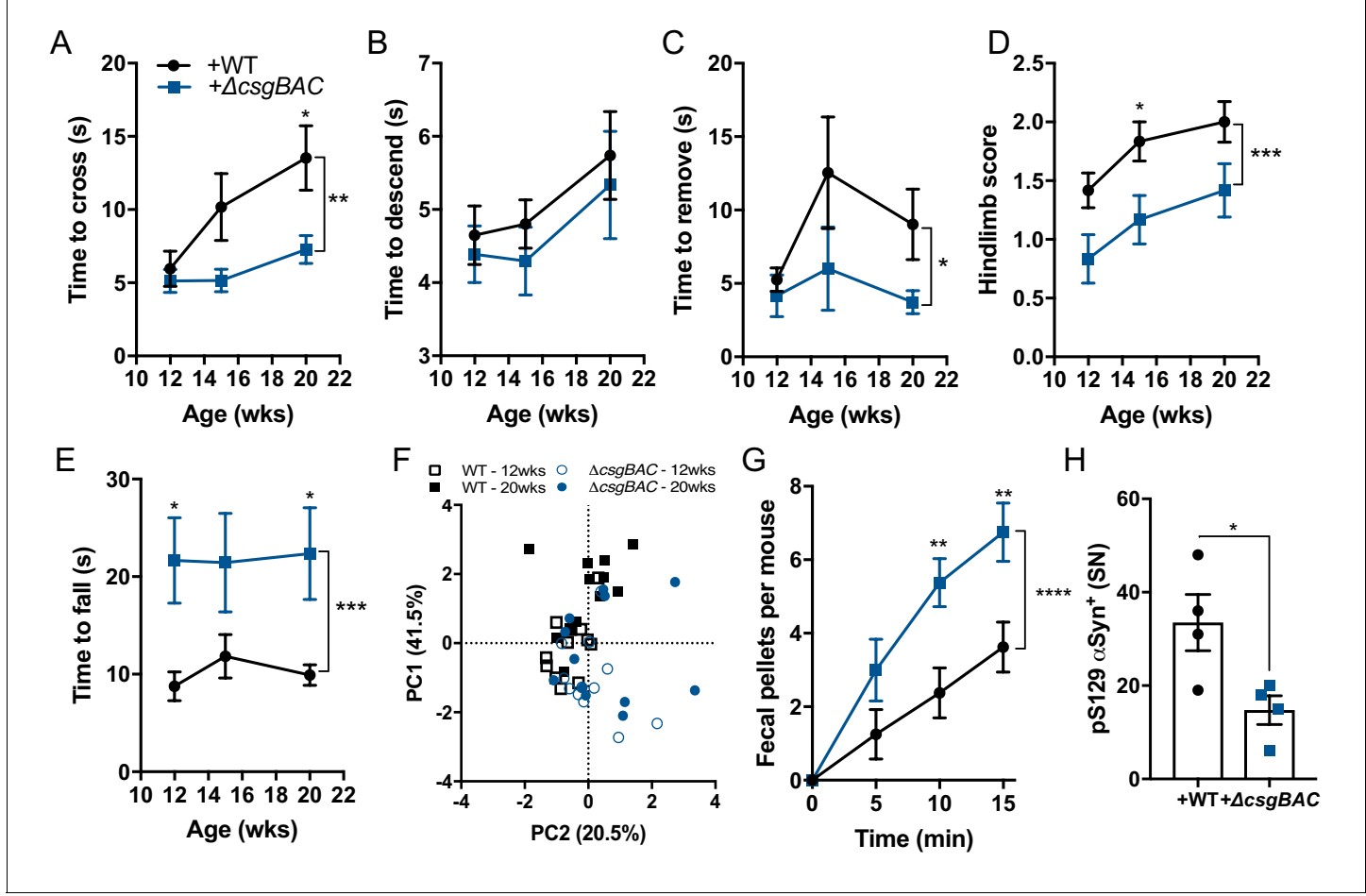

**Figure 2.** Curli biosynthesis within a complex microbiome contributes to motor and GI deficits Germ-free Thy1-αSyn (ASO) mice were colonized with fecal microbes derived from healthy human at 5–6 weeks of age, and concurrently supplemented with either wild-type *E. coli* MC4100 (+WT) or a curli-deficient strain (+ΔcsgBAC). Motor function was tested longitudinally at 12, 15, and 20 weeks of age in the (A) beam traversal, (B) pole descent, (C) adhesive removal, (D) hindlimb clasping score, (E) wirehang tests. (F) Principal component analysis of compiled motor scores from tests in (A–E). (G) Fecal output over a 15-min period observed at week 21 of age. Quantification of (H) pS129αSyn by immunofluorescence microscopy in the substantia nigra. n = 12 (A–F), n = 6 (G), n = 4 (H). Data points represent individuals, bars represent the mean and standard error. Time courses analyzed by two-way ANOVA, with Sidak post-hoc test for between group comparisons indicated above individual time points, and brackets indicating significance between colonization status. Data in (H) analyzed two-tailed *t*-test. *p≤0.05; **p≤0.01; ***p≤0.001; ****p≤0.0001. Motor data are compiled from two independent cohorts.

The online version of this article includes the following source data and figure supplement(s) for figure 2:

**Source data 1.** Source data and statistical analysis.

**Figure supplement 1.** Amyloid-producing bacteria in humanized animals modulate microglia responses.

**Figure supplement 1—source data 1.** Source data and statistical analysis.

indicative of inflammatory status in the midbrain (*Figure 2—figure supplement 1K–N*). Therefore, in the context of a complex human microbiota, curli-producing *E. coli* are sufficient to modulate αSyn-mediated pathophysiology.

## The bacterial amyloid protein, CsgA, accelerates αSyn fibrilization

Emerging data suggest that non-orthologous amyloid proteins can accelerate heterologous aggregation (*Clinton et al., 2010*; *Brettschneider et al., 2015*; *Katorcha et al., 2017*). We therefore tested whether CsgA can directly impact aggregation of αSyn. Purified monomeric CsgA accelerated production of αSyn aggregates during in vitro biochemical amyloid assays (*Figure 3A–C* and *Figure 3—figure supplement 1A*), even at concentrations below which CsgA self-aggregates

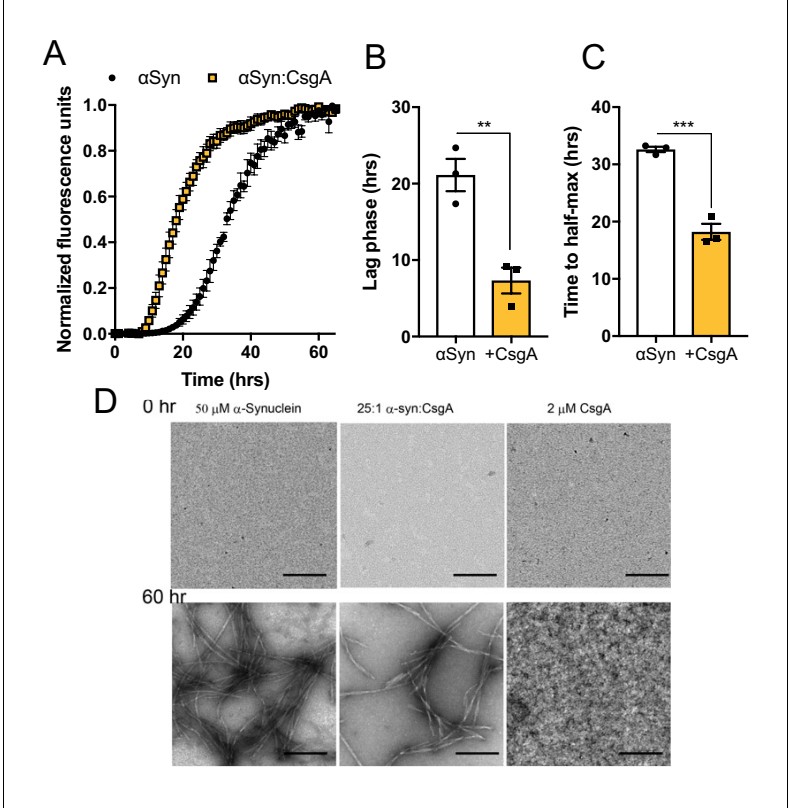

**Figure 3.** The bacterial amyloid protein, CsgA, accelerates αSyn fibrilization In vitro biophysical analysis with purified αSyn and CsgA proteins. (A) Aggregation as measured by Thioflavin T fluorescence over time during αSyn amyloid formation alone or in the presence of CsgA monomers (25:1 molar ratio). Time to reach (B) exponential fibrilization/lag phase and (C) half-maximum from reactions in (A). (D) Representative transmission electron micrographs of αSyn or CsgA alone, or in combination, at 0 and 60 hr post-aggregation. n = 3 (A–C). Bars represent the mean and standard error. Data are analyzed by two-tailed, *t*-test. **p≤0.01; ***p≤0.001.
The online version of this article includes the following source data and figure supplement(s) for figure 3:

**Source data 1.** Source data and statistical analysis.
**Figure supplement 1.** CsgA seeds αSyn aggregation and propagation in vitro.
**Figure supplement 1—source data 1.** Source data and statistical analysis.
**Figure supplement 2.** CsgA seeds αSyn aggregation in vitro and in vivo through native amyloidogenic properties.
**Figure supplement 2—source data 1.** Source data and statistical analysis.
**Figure supplement 3.** Intestinal injection of amyloidogenic curli promotes progressive motor dysfunction.
**Figure supplement 3—source data 1.** Source data and statistical analysis.

(*Figure 3D*). The αSyn fibrils purified from these reactions maintained an ability to accelerate αSyn amyloidogenesis (*Figure 3—figure supplement 1B,C*), suggesting that CsgA-induced αSyn aggregates are propagation-competent, similar to other amyloids (*Spires-Jones et al., 2017*). Surprisingly, we could not detect a direct interaction between purified CsgA and αSyn monomers by surface plasmon resonance (*Figure 3—figure supplement 1D*), perhaps indicating transient interactions or requirements for prior oligomerization of either or both proteins. Unlike αSyn, Tau aggregation was not accelerated by CsgA (*Figure 3—figure supplement 1E–G*). Native amyloid-forming properties of CsgA are required, as a non-amyloidogenic form of CsgA (CsgA:Q49A/N54A/Q139A/N144A; 'SlowGo' *Wang and Chapman, 2008*) did not augment αSyn aggregation in vitro (*Figure 3—figure supplement 2A–C*). To investigate the dependence of these amyloidogenic residues in vivo, we mono-colonized ASO mice with *E. coli* producing either wild-type CsgA or CsgA:SlowGo proteins (*Wang and Chapman, 2008*). Motor performance in ASO mice harboring SlowGo-producing *E. coli* was less severe in comparison to mice colonized with WT bacteria (*Figure 3—figure supplement 2D–H*), although not to the extent of the ΔcsgBAC strain, qualitatively (see *Figure 1*).

Next, a peptide spanning the aggregation domain of CsgA or a non-amyloidogenic version (N122A) (*Tükel et al., 2009*) were injected directly into the intestinal wall of SPF ASO mice. Intraintestinal delivery of amyloidogenic CsgA peptide, but not the mutant peptide, resulted in progressive motor and GI dysfunction (*Figure 3—figure supplement 3A–G*). Furthermore, increased αSyn fibrils were detected in the midbrains of amyloidogenic peptide-injected animals (*Figure 3—figure supplement 3H*). We conclude that gut exposure to a bacterial amyloid is sufficient to exacerbate motor deficits and αSyn brain pathology in this mouse model, in a manner dependent on CsgA aggregation.

## Curli-driven pathophysiology in mice requires functional amyloid formation

Epigallocatechin gallate (EGCG) is a plant-derived, dietary polyphenol that physically inhibits amyloid formation, including αSyn aggregation (*Bieschke et al., 2010*). EGCG is also capable of blocking CsgA amyloidogenesis and represses *csgA* transcript expression in *E. coli* through activation of specific stress response pathways within the bacterial cell (*Serra et al., 2016*). We reveal here that EGCG treatment did not impair *E. coli* growth in culture, but significantly reduced biofilm formation, a process dependent on the production and assembly of curli (*Figure 4—figure supplement 1A, B*) (*Tursi and Tükel, 2018*). Additionally, EGCG also inhibited CsgA-accelerated αSyn amyloid formation during in vitro biochemical aggregation assays (*Figure 4A*). Notably, EGCG remains largely gut-restricted in rodents and humans and is not readily bioavailable in circulation or brain tissues following oral administration (*Lambert et al., 2003*; *Lin et al., 2007*; *Cai et al., 2018*). Oral treatment of wild-type *E. coli* mono-colonized ASO mice with EGCG did not affect fecal *E. coli* abundance, but resulted in decreased *csgA* production (*Figure 4B* and *Figure 4—figure supplement 1C*). Assessment of motor performance revealed that EGCG improved both motor and GI defects exacerbated by curli-producing *E. coli* in ASO mice (*Figure 4C–I* and *Figure 4—figure supplement 1D–I*). In addition, oral EGCG administration reduced αSyn aggregation and pS129-αSyn deposition in both the striatum and midbrain (*Figure 4J* and *Figure 4—figure supplement 1J–M*). Furthermore, microglia morphological changes were limited in EGCG-treated mice (*Figure 4—figure supplement 1O–R*). In addition to its ability to directly inhibit aggregation of bacterial and host-derived amyloids, EGCG has antioxidant and anti-inflammatory activities (*Li et al., 2004*) that may also contribute to the rescue of CsgA-accelerated, αSyn-dependent pathophysiology. Collectively, these data reveal that curli is a specific bacterial structure that can accelerate mammalian amyloid aggregation in vitro and in vivo.

## Conclusion

The majority of synucleinopathy incidences are idiopathic, with multifactorial and complex risks that contribute to disease initiation and/or progression (*Ritz et al., 2016*; *Soldner et al., 2016*; *Johnson et al., 2019*). Our findings reveal that the bacterial amyloid CsgA can accelerate αSyn aggregation and enhance motor abnormalities in mice that are genetically predisposed to αSyn pathology. Curli production in the gut of non-susceptible mice does not impair motor performance at the timepoints examined, suggesting that this microbial trigger is not sufficient, but rather requires additional predisposing factors to promote disease outcomes. Further, inhibiting amyloid formation in the gut correlates with improvements in curli-induced behaviors and pathology. Our study does not reconcile how curli production within the gastrointestinal tract ultimately manifests αSyn aggregation in the brain. Experimental evidence exists for a prion-like spread of αSyn aggregates (*Woerman et al., 2015*), including propagation from the gut to the midbrain via the vagus nerve and/or spinal cord in rodent models (*Uemura et al., 2018*; *Kim et al., 2019*; *Van Den Berge et al., 2019*), although evidence for sustained propagation from the GI tract to structures outside the brainstem in primates is currently lacking (*Manfredsson et al., 2018*). Epidemiological studies suggest an association between full truncal vagotomy with a decreased risk of PD (*Svensson et al., 2015*). Intriguingly, there appears to be a positive correlation between inflammatory bowel disease and neurodegenerative disease (*Hui et al., 2018*; *Peter et al., 2018*). In experimental models, induction of GI inflammation, for instance through LPS or dextran sodium sulfate administration, is sufficient to induce αSyn pathology in the CNS (*Choi et al., 2018*; *Kishimoto et al., 2019*; *Perez-Pardo et al., 2019*). Aggregates of both CsgA and αSyn are capable of signaling through the innate

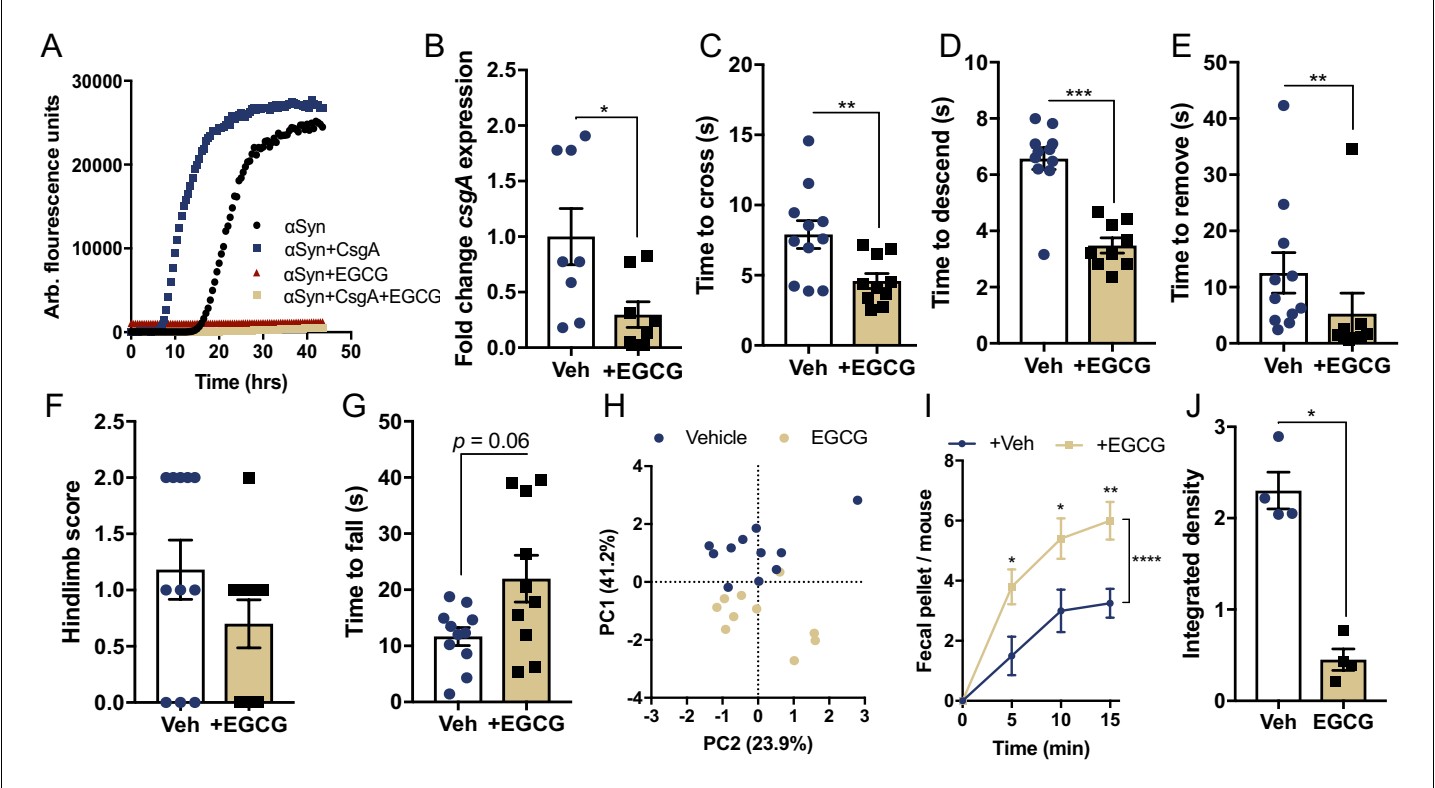

**Figure 4.** Curli-driven pathophysiology in mice requires functional amyloid formation. (**A**), Representative in vitro αSyn aggregation measured by Thioflavin T fluorescence during αSyn amyloid formation alone or in the presence of CsgA (25:1 molar ratio), with and without EGCG (50 μM) treatment. (**B–H**) Germ-free Thy1-αSyn mice (ASO) were mono-colonized with WT *E. coli* at 5–6 weeks of age, and given water alone (Vehicle: Veh) or treated with EGCG *ad lib* in drinking water (+EGCG). (**B**) RNA was extracted from fecal pellets and *csgA* expression quantified by qRT-PCR, relative to *rrsA*. Motor function was assessed at 15–16 weeks of age by quantifying (**C**) beam traversal time, (**D**) pole descent time, (**E**) nasal adhesive removal time, (**F**) hindlimb clasping score, (**G**) wirehang tests. (**H**) Principal component analysis of compiled motor scores from tests in (**C–G**). (**I**) Fecal output over a 15 min period. (**J**) Quantification of insoluble αSyn fibrils in the ventral midbrain by dot blot assay. n = 8 (**B**); n = 10–11 (**C–G**); n = 4–5 (**I, J**). Points represent individuals, bars represent the mean and standard error. Data analyzed by two-tailed Mann-Whitney for **B–G**, (**J**) and two-way ANOVA with Sidek's post-hoc test for **I** with between group comparisons indicated above individual time points, and brackets indicating significance between treatment status. *p≤0.05; **p≤0.01; ***p≤0.001; ****p≤0.0001. Motor data compiled from two independent cohorts.

The online version of this article includes the following source data and figure supplement(s) for figure 4:

**Source data 1.** Source data and statistical analysis.

**Figure supplement 1.** Inhibition of functional amyloid formation dampens progressive pathophysiology.

**Figure supplement 1—source data 1.** Source data and statistical analysis.

immune receptor Toll-like receptor 2 (TLR2) leading to increased inflammatory responses (*Tükel et al., 2009*; *Poewe et al., 2017*). In fact, inhibiting TLR2 in αSyn-based mouse models leads to improved pathology and motor outcomes (*Kim et al., 2015*; *Kim et al., 2018*). Our data demonstrate curli-dependent increases of inflammatory markers in both the intestine and brain, as well as acceleration of αSyn pathology, but do not determine which of these may be the primary driver of curli-mediated pathophysiology. Disentangling these complex relationships in future studies will define how pathologic signals from bacterial amyloids are transduced from the gut to the brain, via transneuronal spread of αSyn aggregates and/or an inflammatory cascade, and mechanisms of gut-to-brain signaling that contribute to motor impairment.

Synucleinopathy is linked to marked changes to the structure of the microbiome. Numerous studies have reported microbiome differences in persons with PD compared to unaffected individuals (*Hasegawa et al., 2015*; *Keshavarzian et al., 2015*; *Scheperjans et al., 2015*; *Unger et al., 2016*; *Bedarf et al., 2017*; *Hill-Burns et al., 2017*; *Hopfner et al., 2017*; *Li et al., 2017*; *Barichella et al., 2019*; *Lin et al., 2018*); however, examination of the gut microbiome during other

synucleinopathies, such as MSA or LBD are still in their infancy (*Engen et al., 2017*). Some reports have identified the family Enterobacteriaceae as enriched in PD compared to unaffected individuals (*Unger et al., 2016*; *Li et al., 2017*; *Barichella et al., 2019*). In those studies which performed correlation analysis between disease severity and microbiome composition, enrichment of Enterobacteriaceae is observed to be positively associated with worsened motor symptoms (*Scheperjans et al., 2015*; *Li et al., 2017*). To our knowledge, no study to date has identified differential abundance of *csgA* or curli-encoding genes specifically in human incidences, and only a single study to date has utilized metagenomic sequencing of the gut microbiome during PD (*Bedarf et al., 2017*). Despite recent advances defining microbiome changes in neurodegenerative diseases, a contributing role by gut bacteria to synucleinopathies in humans remains correlative (reviewed in *Sampson, 2019*). In experimental models, however, transplant of fecal microbiomes from PD patients into ASO mice results in greater motor deficits than microbiomes from healthy controls (*Sampson et al., 2016*), suggesting functional consequences to changes in gut microbial composition.

Although our findings herein are limited to a single transgenic animal mouse model, and similar pathologies are observed in aged rats and nematodes (*Chen et al., 2016*), it is possible that these observations are model specific. Non-amyloid contributions from diverse gut bacteria are also likely to occur and influence neurodegenerative outcomes. For instance, enrichment of the gut bacterium *Proteus mirabilis*, or intestinal administration of its purified LPS, impairs the dopaminergic system of mice and increases susceptibility to neurotoxins (*Choi et al., 2018*). Infection with Gram-negative bacteria, including *Citrobacter* sp. (a taxonomic relative of *E. coli*), can trigger a pathological process leading to neurodegenerative immune responses in the brain and loss of midbrain dopamine production in a PD-relevant mouse model (*Matheoud et al., 2019*). In addition, specific microbiome-derived metabolites promote microglia maturation (*Erny et al., 2015*) and enhance αSyn-dependent pathology in germ-free mice (*Sampson et al., 2016*). Conversely, specific microbial metabolites (*Blacher et al., 2019*) or fecal microbiome transplants (*Sun et al., 2018*) may provide protection from neurodegenerative insults.

While CsgA is one example of bacterially-produced amyloid, microbial amyloid formation in general may influence physiological processes with outcomes relevant for neurodegenerative disease (*Friedland and Chapman, 2017*). For example, the *Pseudomonas* sp. functional amyloid, FapC, may also accelerate αSyn aggregation in vitro (*Christensen et al., 2019*), and bacterial chaperones of CsgA modulate αSyn amyloidogenesis (*Evans et al., 2015*). More broadly, CsgA is observed to accelerate amyloid β aggregation in vitro (*Hartman et al., 2013*; *Perov et al., 2019*), as well as the disease-associated amyloids, islet amyloid polypeptide (IAPP) and semen enhancer of viral infection (SEVI) in vitro (*Hartman et al., 2013*), and serum amyloid A in mice (*Lundmark et al., 2005*). Emerging evidence in diverse animal models suggests that gut bacteria may modulate amyloid-driven diseases, not only in synucleinopathy as we and others describe, but Alzheimer's disease (*Harach et al., 2017*; *Dodiya et al., 2019*) and amyotrophic lateral sclerosis (ALS) as well (*Blacher et al., 2019*), providing justification for future human studies and revealing possible new targets for interventions that may prevent, slow, or halt amyloid formation and neurodegenerative disease.

# Materials and methods

## Key resources table

| Reagent type (species) or resource | Designation | Source or reference | Identifiers | Additional information |
|---|---|---|---|---|
| Strain, strain background (*Mus musculus*) | Thy1- alpha-synuclein (Line 61), BDF1 background | *Rockenstein et al., 2002* | ASO (alpha-synuclein over-expressing) | With permission, UCSD |
| Strain, strain background (*Escherichia coli*) | Str. K12, MC4100 | *Zhou et al., 2013* | WT (wild-type) | |

*Continued on next page*

*Continued*

| Reagent type (species) or resource | Designation | Source or reference | Identifiers | Additional information |
|---|---|---|---|---|
| Strain, strain background (*Escherichia coli*) | Str. K12, MC4100:Δ*csgBAC* | *Zhou et al., 2013* | Δ*csgBAC* | |
| Strain (*Escherichia coli*) | Str. K12, MC4100: CsgA:Q49A/N54A/ Q139A/N144A | *Wang and Chapman, 2008* | SlowGo | |
| Antibody | Anti-alpha synuclein, mouse monoclonal | BD | Cat#: 610787 | (1:1000) |
| Antibody | Anti-pS129 alpha-synuclein, rabbit monoclonal | AbCam | Cat#: Ab51253 | (1:1000) |
| Antibody | Anti-tyrosine hydroxylase, mouse monoclonal | Millipore | Cat#: MAB318 | (1:1000) |
| Antibody | Anti-Iba1, rabbit | Wako | Cat#: 019–19741 | (1:1000) |
| Antibody | Anti-aggregated alpha-synuclein, rabbit polyclonal | AbCam | Cat#: MJFR-14-6-4-2 | (1:1000) |
| Antibody | Anti-mouse IgG-546, goat polyclonal | Life Technologies | Cat#: A-11003 | (1:1000) |
| Antibody | Anti-rabbit IgG-546, goat polyclonal | Life Technologies | Cat#: A-11010 | (1:1000) |
| Antibody | Anti-mouse IgG-488, goat polyclonal | Life Technologies | Cat#: A-11001 | (1:1000) |
| Antibody | Anti-rabbit IgG-HRP, goat polyclonal | Cell Signaling | Cat#: 7074 | (1:1000) |
| Peptide, recombinant protein | CsgA (N'-QYGGNN-C') | Bio-Synthesis, Inc; *Tükel et al., 2009* | CsgA | |
| Peptide, recombinant protein | N122A (N'QYGGNA-C') | Bio-Synthesis, Inc; *Tükel et al., 2009* | N122A | |
| Sequence-based reagent | 16S rRNA: 5'-TCCTACGGGAG GCAGCAGT-3' and 5'-GGACTACCAGGG TATCTAATCCTGTT-3' | IDT | qPCR primer | PrimerBank |
| Sequence-based reagent | *rrsA:* 5'-AGTGATAAACTG GAGGAGGTG-3' and 5'-GGACTACGACG CACTTTATGAG-3' | IDT | qPCR primer | PrimerBank |
| Sequence-based reagent | *csgA:* 5'-ATGACGGTTAA ACAGTTCGG-3' and 5'-AGGAGTTAGAT GCAGTCTGG-3' | IDT | qPCR primer | PrimerBank |
| Sequence-based reagent | *gapdh:* 5'-CATGGCCTTCC GTGTTCCTA-3' and 5'-CCTGCTTCACCA CCTTCTTGAT-3' | IDT | qPCR primer | PrimerBank |

*Continued on next page*

*Continued*

| Reagent type (species) or resource | Designation | Source or reference | Identifiers | Additional information |
|---|---|---|---|---|
| Sequence-based reagent | *il6*: 5'-TAGTCCTTCCTA CCCCAATTTCC-3' and 5'-TTGGTCCTTAGC CACTCCTTC-3', | IDT | qPCR primer | PrimerBank |
| Sequence-based reagent | *TH*: 5'-CCAAGGTTCAT TGGACGGC-3' and 5'-CTCTCCTCGAAT ACCACAGCC-3' | IDT | qPCR primer | PrimerBank |
| Sequence-based reagent | *tnfa*; 5'-CCCTCACACTCA GATCATCTTCT-3' and 5'-GCTACGACAG TGGGCTACAG-3' | IDT | qPCR primer | PrimerBank |

## Animals

Male wild-type and Thy1-αSyn mice (Line 61, with permission from University of California, San Diego) were generated as described previously (*Rockenstein et al., 2002*; *Chesselet et al., 2012*; *Sampson et al., 2016*), through breeding female BDF1 background Thy1-αSyn mice to male BDF1 offspring generated via crossing female C57BL/6 with DBA/2 males (Charles River, Hollister, CA), and refreshing breeding pairs ~ 6 months. Germ-free (GF) Thy1-αSyn breeding pairs were generated via caesarian section and cross-fostered by GF Swiss-Webster dams. Conventionally colonized and mono-colonized animals were housed in autoclaved, ventilated, microisolator caging. GF animals were housed in open-top caging within flexible film isolators and maintained microbiologically sterile, confirmed via 16S rRNA PCR from fecal-derived DNA and culture of fecal pellets on Brucella blood agar or tryptic soy blood agar (Teknova, Hollister, CA) under anaerobic and aerobic conditions, respectively. Mono-colonized animals received ~$10^8$ cfu of the indicated bacterial strains in ~100 µL sodium bicarbonate buffer (5% w/v) at 5–6 weeks of age. Human microbiome colonized animals received ~100 µL 0.1 g/mL fecal extract from previously sampled healthy human donor (ENA Accession: PRJEB17694; Sample #: 10483.donor2.HC; MMA_008 [*Sampson et al., 2016*], California Institute of Technology's Institutional Review Board #15–0568- exempt), at 5–6 weeks of age. For mice mono-colonized with *E. coli* containing plasmid vectors for CsgA and CsgA:SlowGo, drinking water was supplemented with 50 µg/mL kanamycin (Sigma Aldrich, St Louis, MO). Human microbiome colonized mice were associated with WT or Δ*csgBAC* mutant *E. coli* at time of microbiota transplant. For epigallocatechin gallate (EGCG) treatment, animals received filter-sterilized EGCG (Sigma Aldrich) at 200 µg/mL in drinking water ad libitum beginning at 5–6 weeks of age. All animals received autoclaved food (LabDiet Laboratory Autoclavable Diet 5010, St Louis, MO) and water ad libitum, were maintained on the same 12 hr light-dark cycle, and housed in the same room of the facility. All animal husbandry and experiments were approved by the California Institute of Technology's Institutional Animal Care and Use Committee (IACUC), through protocol #1707.

## Motor function assessment

Motor function was assessed similarly to previous studies (*Fleming et al., 2004*; *Sampson et al., 2016*), between hours 7 and 9 of the light phase, within a biosafety cabinet. Beam traversal and pole descent were performed first, followed by fecal output measurement. One day later, wirehang, adhesive removal and hindlimb scoring was performed. *Beam traversal-* A 1 m plexiglass beam consisting of four segments of 0.25 m in length (Stark's Plastics, Forest Park, OH) was constructed with consecutively thinner widths of 3.5, 2.5, 1.5, and 0.5 cm, with 1 cm overhangs placed 1 cm below the surface of the beam and the narrowest end placed into home cage. Animals were trained for two consecutive days before the first testing. On the first day of training, animals received one trial with the home cage positioned close to the widest segment, where the animal was placed, and guided the animals forward along the narrowing beam. Animals received two more trials with limited or no assistance to encourage forward movement and stability on the beam. On the second day of

training, animals were given three trials to traverse the beam, with little to no assistance. On the third day, animals were timed over three trials to traverse from the widest segment to the home cage. Time was measured from the placement of the animal's forelimbs onto the second segment until a forelimb reached the home cage. Score was averaged over three trials. *Pole descent-* A 0.5 m long pole, 1 cm in diameter, wrapped with non-adhesive shelf liner to facilitate the animals grip, was placed into the home cage, with animals removed and placed into fresh housing. Animals received 2 days of training to descend from the top of the pole and into the home cage. On the first day, animals received three trials to descend the pole. First animals were placed head-down ~1/3 the distance above the base, the second trial from ~2/3 the height, and on the third trial animals were placed at the top of the pole, head-down. The second day of training, animals were tasked with descending, head-down, from the top of the pole, three times. On the day of testing, animals were placed head-down on the top of the pole and timed beginning when the experimenter released the animal and ended when one hind-limb was placed on the pole base. Score was averaged over three trials. *Adhesive removal-* ¼" round adhesive labels (Avery, Glendale, CA) were placed on the nasal bridge. Animals were placed into their home cage (with cage mates removed into separate cage) and timed to completely remove the sticker. Animals were recorded over two trials, and averaged. *Hindlimb clasping reflex scoring-* Animals were gently lifted by the mid-section of the tail and observed over ~5–10 s (*Zhang et al., 2014*). Animals were assigned a score of 0, 1, 2, or 3. 0 was scored to animals that freely moved both their limbs and extended them outwards. A 1 was assigned to animals which clasped one hindlimb inward for the duration of the restraint or if both legs exhibited partial inward clasping. 2 was given if both legs clasped inward for the majority of the observation, but still exhibited some flexibility. A score of 3 was assigned if animals displayed complete rigidity of hindlimbs that immediately clasped inward and exhibited no signs of flexibility. For animals in *Figure 3—figure supplement 2G*, animals were scored twice on 2 consecutive days and the score averaged, the remainder were scored once. *Wirehang-* Animals were placed in the center of a 30 cm by 30 cm screen with 1 cm wide mesh. The screen was inverted head-over-tail and placed on supports ~ 40 cm above an open cage with deep bedding. Animals were timed until they released their grip or remained for 60 s, and the score from two trials averaged. *Fecal Output-* Animals were removed from their home cages and placed into a 12 cm x 25 cm translucent cylinder. Fecal pellets were counted every 5 min, cumulative over 15 min. *Principal component analysis-* PCA was performed using ClustVis web tool, with default settings and reversed axis for display (*Metsalu and Vilo, 2015*).

## Bacterial strains, manipulations, and characterizations

*Escherichia coli* K12 str. MC4100 and the previously characterized isogenic deletion mutant of the *csgBAC* operon were cultured aerobically in YESCA media at 37°C (*Zhou et al., 2013*). Biofilm assays were performed via crystal violet staining of static culture at room temperature in YESCA as described previously (*Zhou et al., 2013*). Congo red staining was performed on YESCA agar following 2–3 days growth at room temperature as described previously (*Zhou et al., 2013*). Fecal bacterial DNA was isolated using QuickDNA Fecal/Soil Microbe Miniprep (Zymo Research, Irvine, CA). Fecal bacterial RNA was isolated using PowerMicrobiome RNA Isolation kit (MoBio, Carlsbad, CA) and cDNA generated via iScript cDNA Synthesis kit (BioRad, Hercules, CA). qPCR was performed with SybrGreen master mix (Applied Biosystems, Foster City, CA) on an AB7900ht instrument using the following primers: 16 s rRNA-5'-TCCTACGGGAGGCAGCAGT-3' and 5'-GGACTACCAGGGTATCTAATCCTGTT-3'; *rrsA*-5'-AGTGATAAACTGGAGGAGGTG- 'and 5'-GGACTACGACGCACTTTATGAG-3'; *csgA*- 5'-ATGACGGTTAAACAGTTCGG-3' and 5'-AGGAGTTAGATGCAGTCTGG-3'. *Bacteroides fragilis* str. NTCT9343 was cultured anaerobically in brain-heart infusion (BHI) broth at 37°C, and colonized into GF animals via oral gavage of ~$10^8$ cfu in 100 µL sodium bicarbonate buffer (5% w/v) at 5–6 weeks of age. Segmented filamentous bacteria were colonized via bedding transfer and co-housing within a mono-associated gnotobiotic isolator. Colonization was confirmed by PCR for SFB 16 s RNA. For growth curves, *E. coli* was first grown aerobically at 37°C overnight in BHI broth, and subcultured at 1:400 in BHI containing indicated concentrations of EGCG, in a 200 µL volume in a 96-well plate. Plates were incubated at 37°C aerobically, with orbital shaking in a BioTek Cytation 5 plate reader, and the optical density at 600 nm measured every hour. Endotoxin content was measured following overnight growth at 37°C, aerobically in BHI using the Pierce LAL Chromogenic Endotoxin kit (ThermoFisher, Pittsburgh, PA) according to manufacturer's instructions. LPS was

stained from lysates derived from identically grown bacterial cultures following separation on a 4–20% SDS-PAGE gel with Pro-Q Emerald 300 LPS stain kit (ThermoFisher), according to manufacturer's instructions.

## α-Synuclein aggregation assays

*Thioflavin T (ThT) assays*-Freshly purified CsgA (both wild-type and 'SlowGo') and αSyn were diluted in 50 mM KPi (pH 7.3) to the molar concentrations indicated in each experiment. Samples were incubated in 96-well, black, flat bottom plates at 37°C with 20 µM ThT and 100 mM NaCl under continuous shaking conditions, along with a 1 mm glass bead for homogenous mixing. The ThT fluorescence intensity was recorded in 30 min interval using a Tecan plate reader (excitation: 438 nm; emission: 495 nm; cut-off: 475 nm). *TauK18 aggregation assay*- 50 µM of TauK18 fragment with heparin (TauK18:Heparin 4:1) was incubated with or without CsgA (2 µM) in 14 mM MES buffer pH 6.8 at 37°C with continuous shaking in 96-well, black, flat bottom plates. 2 mm beads and 20 µM ThT was added to all the wells. The ThT fluorescence was monitored with previously mentioned parameters after 30 min in FLUOstar omega plate reader (BMG Labtech). *Transmission electron microscopy*- Aliquots of CsgA and αSyn reactions were taken at indicated timepoints. Five microliters of sample was applied on glow-discharged carbon-coated grids, incubated for 1 min and washed with MilliQ water before staining with 1% uranyl acetate. Samples were imaged on Jeol electron microscope (JEOL-1400 plus). *Circular dichroism spectroscopy*- αSyn alone or with CsgA (in 20 mM KPi pH 7.3) were analyzed using a Jasco J-810 spectropolarimeter from 190 nm to 250 nm at 25°C immediately after purification and at indicated timepoints.

## Synuclein pathology and inflammatory responses

*CD11b enrichment*- Animals were sedated with pentobarbital and perfused; whole brains were homogenized in PBS via passage through a 100 µm mesh filter, myelin debris were removed using magnetic separation with Myelin Removal Beads (Miltenyi Biotec, San Diego, CA), and subsequently CD11b positive enrichment performed similarly, via Microglia Microbeads (Miltenyi Biotec), according to manufacturer's instructions. Cells were immediately lysed in Trizol and RNA extracted with the DirectZol RNA extraction (Zymo Research). cDNA and qPCR performed as described above with primers- *gapdh*: 5'-CATGGCCTTCCGTGTTCCTA-3' and 5'-CCTGCTTCACCACCTTCTTGAT-3'; *il6*: 5'-TAGTCCTTCCTACCCCAATTTCC-3' and 5'-TTGGTCCTTAGCCACTCCTTC-3', *tyrosine hydroxylase*: 5'-CCAAGGTTCATTGGACGGC-3' and 5'-CTCTCCTCGAATACCACAGCC-3'; and *tnfa*; 5'-CCCTCACACTCAGATCATCTTCT-3' and 5'-gctacgacagtgggctacag-3'. *Synuclein imaging*- Perfused whole brains were dissected and fixed with 4% (w/v) paraformaldehyde. 50 µm sections were generated via vibratome. For proteinase-K resistant staining, free-floating sections were treated briefly with 5 µg/mL proteinase K (NEB, Ipswich, MA) and stained with anti-alpha synuclein (1:1000 mouse; #610787 BD, Franklin Lakes, NJ), pS129-synuclein staining utilized anti-pS129syn (1:1000 rabbit; #ab51253 AbCam, Cambridge, UK) and anti-tyrosine hydroxylase (1:1000 mouse; MAB318, Millipore, Burlington, MA). Sections were stained with secondary anti-mouse IgG-AF488 (1:1000, Life Technologies, Carlsbad, CA), anti-rabbit IgG-546 (1:1000, Life Technologies) and Neurotrace (Life Technologies), mounted with ProFade Diamond DAPI (Life Technologies) and imaged with a 20x objective on a Zeiss LSM800 confocal microscope. Sections corresponding to ~1500 µm from midline were counted manually for $TH^+$ cell bodies and pS129syn puncta in ImageJ software. *Microglia reconstructions*- Microglia were imaged and analyzed as previously described (*Erny et al., 2015*; *Sampson et al., 2016*). Thin sections prepared as above were stained with anti-Iba1 (1:1000 rabbit, Wako, Richmond, VA) and anti-rabbit IgG-AF546 (1:1000, Life Technologies). Z-stacks were imaged at 1 µm steps and analyzed using Imaris software. *ELISAs*- For TNFα and IL6 (eBioscience, San Diego, CA), αSyn (ThermoFisher), and dopamine (Eagle Biosciences, Nashua, NH) tissue homogenates were prepared in RIPA buffer containing protease inhibitor (ThermoFisher), and ELISA performed according to manufacturer's instructions with 100 µg of tissue. Multiplexed intestine (100 µg) and serum cytokine levels were measured on a Bio-Plex 200 using the Bio-Plex Mouse Cytokine 23-plex Assay (Biorad) according to manufacturer's instructions. *Dot blot*- Tissue homogenates were spotted onto 0.45 µm nitrocellulose membranes, in 1 µg/µL aliquots. Membranes were blocked with 5% dry skim milk in Tris-buffered saline and stained with anti-aggregated alpha-synuclein antibody (1:1000, MJFR-14-6-4-2, rabbit, AbCam) and anti-rabbit IgG-HRP (1:1000, Cell Signaling Technology,

Danvers, MA), and detected with Clarity chemiluminescence substrate (BioRad) on a BioRad GelDoc XR and densitometry performed.

## Intestinal injections

Synthetic hexapeptides of CsgA (Nterm-QYGGNN-Cterm) and the non-amyloidogenic mutant N122A (Nterm-QYGGNA-Cterm) were purchased from Bio-Synthesis Inc (Lewisville, TX). Mice were anesthetized with isoflurane (1–4% vol/vol) and kept on a self-regulating heating pad throughout the procedure. A 10 μl Hamilton syringe fitted with a 36-gauge beveled needle was loaded with 30 μg of either hexapeptide in saline (9 μl total). Each animal was injected in three locations (both sides of pyloric antrum and duodenum [0.5 cm past the pyloric sphincter]; 3 μl each location) by inserting the needle tip bevel facing up into the intestine wall at a 15° angle. After injection, the abdominal muscle/peritoneal layer and skin were sutured separately, then mice returned to home cages. Mice were injected subcutaneously with buprenorphine SR (1 mg/kg) and monitored for 3 days for normal food and water consumption.

## Statistics and data availability

Data were analyzed in GraphPad Prism, with the analysis indicated in each figure legend. Statistical output and numerical source data presented within the manuscript are available within the supplementary information included with this manuscript.

## Acknowledgements

We thank Drs. E Hsiao, M Sampson, S Campbell, D Yamashita, D Donabedian, and members of the SKM laboratory for helpful critiques and review of this manuscript. We are grateful to J Paramo, S Estrada, K Ly and the Caltech OLAR staff for animal care husbandry, and Y Garcia-Flores (Caltech) for technical support. Electron micrographs of amyloids were generated at the Microscopy and Imaging Analysis Laboratory Core at the University of Michigan. Fluorescent imaging and image analysis were performed in the Caltech Biological Imaging Facility, with the support of the Caltech Beckman Institute and the Arnold and Mabel Beckman Foundation. TRS was a Larry L Hillblom Foundation postdoctoral fellow. This project was supported by funds from the Heritage Medical Research Institute to VG and SKM; the Knut and Alice Wallenberg Foundation and Swedish Research Council to PW-S; the National Institutes of Health grants AG054101 (CC), GM118651 (MRC) and NS085910 (SKM); Axial Biotherapeutics to SKM; and the Department of Defense grant PD160030 to SKM.

# Additional information

## Competing interests

Timothy R Sampson: has intellectual property pending in relationship to the content of this manuscript, US Patent App. 15/893,456 and 16/302,321. Matthew Chapman: a member of the Scientific Advisory Board of Axial Biotherapeutics. Sarkis K Mazmanian: has financial interest in Axial Biotherapeutics. Has intellectual property pending in relationship to the content of this manuscript, US Patent App. 15/893,456 and 16/302,321. The other authors declare that no competing interests exist.

## Funding

| Funder | Grant reference number | Author |
|---|---|---|
| Larry L. Hillblom Foundation | | Timothy R Sampson |
| Heritage Medical Research Institute | | Viviana Gradinaru<br>Sarkis K Mazmanian |
| Knut och Alice Wallenbergs Stiftelse | | Pernilla Wittung-Stafshede |
| Swedish Research Council | | Pernilla Wittung-Stafshede |
| National Institute on Aging | AG054101 | Collin Challis |

| National Institute of General Medical Sciences | GM118651 | Matthew Chapman |
| National Institute of Neurological Disorders and Stroke | NS085910 | Sarkis K Mazmanian |
| Department of Defense | PD160030 | Sarkis K Mazmanian |
| Axial Biotherapeutics | | Sarkis K Mazmanian |

The funders had no role in study design, data collection and interpretation, or the decision to submit the work for publication.

### Author contributions

Timothy R Sampson, Conceptualization, Investigation, Visualization; Collin Challis, Neha Jain, Anastasiya Moiseyenko, Mark S Ladinsky, Istvan Horvath, Investigation, Visualization; Gauri G Shastri, Brittany D Needham, Investigation; Taren Thron, Resources, Investigation; Justine W Debelius, Stefan Janssen, Formal analysis; Rob Knight, Viviana Gradinaru, Supervision; Pernilla Wittung-Stafshede, Matthew Chapman, Sarkis K Mazmanian, Conceptualization, Supervision

### Author ORCIDs

Timothy R Sampson https://orcid.org/0000-0002-2486-8766
Stefan Janssen https://orcid.org/0000-0003-0955-0589
Pernilla Wittung-Stafshede https://orcid.org/0000-0003-1058-1964
Viviana Gradinaru http://orcid.org/0000-0001-5868-348X
Sarkis K Mazmanian https://orcid.org/0000-0003-2713-1513

### Ethics

Animal experimentation: All animal husbandry and experiments were approved by the California Institute of Technology's Institutional Animal Care and Use Committee (IACUC) under protocol #1707.

### Decision letter and Author response

Decision letter https://doi.org/10.7554/eLife.53111.sa1
Author response https://doi.org/10.7554/eLife.53111.sa2

## Additional files

### Supplementary files

• Transparent reporting form

### Data availability

All data generated or analysed during this study are included in the manuscript and supporting files. Source data files and statistical output for all figures have been provided.

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
