## [Decision Letter]

**Acceptance summary:**

This is an exciting study demonstrating how gut bacteria and associated curli proteins can promote α-synuclein aggregation and pathology both in the gut and in the brain that lead to behavioral deficits in mice. There is increasing interest in how the gut microbiome and pathogens can influence the spread and development of synucleinopathies such as Parkinson's disease. This study opens up questions of whether dysregulation of bacterial strains that produce curli or other similar proteins are associated with synucleinopathies in humans. Thank you for contributing your study.

**Decision letter after peer review:**

Thank you for submitting your article "A gut bacterial amyloid promotes α-synuclein aggregation and motor impairment in mice" for consideration by *eLife*. Your article has been reviewed by three peer reviewers, one of whom is a member of our Board of Reviewing Editors, and the evaluation has been overseen by a Reviewing Editor and Wendy Garrett as the Senior Editor. The following individuals involved in review of your submission have agreed to reveal their identity: Michel Desjardins (Reviewer #3).

The reviewers have discussed the reviews with one another and the Reviewing Editor has drafted this decision to help you prepare a revised submission.

Summary:

In this manuscript, Sampson et al. present evidence that bacteria in the gut can lead to the emergence of motor impairment in a PD-susceptible mouse model through the action of a bacterial amyloid protein. They examined whether a bacterially produced amyloid protein, curli, could contribute to the aggregation of mammalian αSyn, leading to PD-related neurological dysfunction. The manuscript is well written and concise. The authors used a "reductionist" approach and demonstrated that amyloid-producing bacteria induce PD-like symptoms in α-synuclein overexpressing mice (ASO). They then went on to show that the curli proteins (bacterial amyloid) has a similar effect when introduced in the gut. These data further support the emerging concept that peripheral events in the gut contributes to the pathophysiological process leading to PD. The study is well focused and built around complimentary sets of experiments with microorganisms and their protein products. Behavioral analyses, as well as characterization of the modification of dopaminergic neurons in the brain are provided. Their data indicates that motor impairment can occur without a significant loss of TH neurons in the substantia nigra but a decrease of dopamine in the striatum (as shown in a recent article by Matheoud et al., 2019). Finally, they used epigallocatechin gallate (EGCG) to block curli's ability to form biofilms and found that it blocked CsgA- accelerated αSyn formation.

All reviewers were overall enthusiastic about the findings presented in this study. However, they agree that it would be important for the authors to show that the motor impairment in their model can be reversed by L-DOPA treatment. Better clarity also about presented data (immunoblots, ENS analysis) would strengthen the manuscript.

Essential revisions:

1) If the behavioral findings are due to exacerbation of PD-like motor phenotype as the authors propose, then the authors should be able provide behavioral data showing that L-DOPA alleviates or reduces the behavioral abnormalities.

2) There seems to be a paucity of pathologic αSyn in TH neurons and the biochemical evidence is weak. It would help if the authors showed their full-length immunoblots of αSyn from the soluble versus insoluble fractions.

3) For neuropathology in the gut, can there be quantitative analysis of enteric neurons? The myenteric plexus was stained for α-synuclein and PGP9.5, but the numbers of enteric neurons were not quantified. Is there loss of neurons in the proximal large intestine?

4) A better explanation of how curli exacerbates the motor phenotype would help. Is it through enhancing inflammation, affecting DA metabolism or affecting other areas that express high levels of α-synuclein in these transgenic mice? Is there a correlation between aggregate burden or inflammation and behavior?

[Editors' note: further revisions were suggested prior to acceptance, as described below.]

Thank you for resubmitting your work entitled "A gut bacterial amyloid promotes α-synuclein aggregation and motor impairment in mice" for further consideration by *eLife*. Your revised article has been evaluated by Wendy Garrett (Senior Editor) and a Reviewing Editor.

The manuscript has been improved but there are some remaining issues that need to be addressed before acceptance, as outlined below:

All reviewers agree that improvements have been made including additional data for quantification of neurons in the gut myenteric plexus, and revisions to the text.

However, there is a major point that must be addressed with necessary revisions to the text to tone down the relevance of their animal model and findings in relation to PD given the inability to reverse the phenotype with L-DOPA in this model (Point #1). An additional point is raised concerning the full immunoblot (Point #2).

1) The authors cited prior published work that the behavioral deficits in the Thy1-αSyn mice are not responsive to L-DOPA and as such argue that this experiment will not provide useful information. While we agree with the authors that in light of these prior published works, L-DOPA experiments are not likely to provide useful information. However, the fact that L-DOPA does not have an effect on the Thy1-αSyn mice indicates that the authors' claims that behavioral findings are due to exacerbation of PD-like motor phenotypes is specious. Since L-DOPA does not rescue the phenotype, while we understand their arguments, this animal model is not reflective of PD. Yet the authors continue to make statements throughout the revised paper that the motor deficits are consistent with the hallmarks of PD. For instance, the Abstract states "*Escherichia coli* exacerbates hallmark disease features such as motor impairment and αSyn pathology in the gut and brain."

The authors need to therefore take a more balanced approach and present the Thy1-αSyn mice as a model of αSyn pathophysiology with accompanying behavior deficits and they should not make claims regarding motor impairment and the link to PD. In other words, the Thy1-αSyn mice represent a model of αSyn induced behavioral abnormalities only and claims regarding PD should be removed from the paper.

2) We could not easily find the requested full immunoblot of soluble vs. insoluble α-synuclein fractionations. It's currently in an embedded tab of an excel file: is there a way to post this on the actual figure instead of as a source data file (Figure 1—figure supplement 2D)?

---

## [Author Response]

Essential revisions:1) If the behavioral findings are due to exacerbation of PD-like motor phenotype as the authors propose, then the authors should be able provide behavioral data showing that L-DOPA alleviates or reduces the behavioral abnormalities.

In the Thy1-αSyn mouse model, loss of dopamine production and degeneration of dopaminergic neurons occurs at 14 months of age (Chesselet et al., 2012), which is significantly later than the appearance of motor dysfunction (2-4 months of age; Fleming et al., 2004), neuroinflammation (1 month of age; Watson et al., 2012), and non-motor symptoms (including gastrointestinal motility at 2-3 months of age; Wang et al., 2012). As described in our study and previous reports (Chesselet et al., 2012; Lam et al., 2011), there is no notable loss of striatal dopamine (Figure 1—figure supplement 2Q) or loss of tyrosine hydroxylase expression (Figure 1—figure supplement 2R) at the time points used throughout our work. Therefore, our data are in complete agreement with published literature for a lack of dopamine dependence at early stages of disease, strongly suggest that dopamine replacement therapy would not have benefits in mice at the ages tested. In fact, this experiment has been performed in SPF mice, which are somewhat different than our colonization paradigms, showing that L-DOPA treatment paradoxically results in slightly increased motor impairment in the Thy1-αSyn mouse model at early ages (Fleming et al., 2006). The reason for this previous observation still remains unknown.

While we are technically able to perform the L-DOPA treatment studies and had previously considered them, the experiment would take over 14 months to complete and we feel it would not provide useful information (based on the published data referenced above and our observations in Figure 1—figure supplement 2Q, R). Accordingly, we make no claims about the role dopamine on motor phenotypes, as the focus of our study is identification of a bacterial pathway that can promote αSyn aggregation and pathophysiology in Thy1-αSyn mice, rather than a better understanding of the mouse model itself.

Aggregation of αSyn is central to both Parkinson’s disease and the pathophysiology of this mouse model. Prior studies have modulated αSyn aggregation through small molecules in this mouse model and have demonstrated motor phenotypes can be limited when αSyn is inhibited (Richter et al., 2017; Wrasidlo et al., 2016), or enhanced when αSyn aggregation is accelerated in vivo (Pokrzywa et al., 2017). Exploring the molecular pathways into how aggregation leads to motor phenotypes would require a separate dedicated study.

We hope the referees and editors agree that the investment in time and effort for L-DOPA treatment experiments outweigh their expected value to the current study. We have done our best to not over-interpret our findings and have restricted discussion to results directly supported by data, namely the discovery that curli from the gut microbiome are sufficient to induce αSyn aggregation and motor phenotypes in the widely-used Thy1-αSyn model of synucleinopathy. In future studies, we plan to test whether curli play a role in other mouse models of disease.

2) There seems to be a paucity of pathologic αSyn in TH neurons and the biochemical evidence is weak. It would help if the authors showed their full-length immunoblots of αSyn from the soluble versus insoluble fractions.

We have included the requested full immunoblot of soluble vs. insoluble α-synuclein fractionations (Figure 1—figure supplement 2D) in Figure 1—source data 1. Thank you for this suggestion.

3) For neuropathology in the gut, can there be quantitative analysis of enteric neurons? The myenteric plexus was stained for α-synuclein and PGP9.5, but the numbers of enteric neurons were not quantified. Is there loss of neurons in the proximal large intestine?

This has been previously investigated, and the Thy1-αSyn mouse does not show degeneration of any neurons in the myenteric plexus (Wang et al., 2012). Similarly, in our hands, we do not observe striking differences in the number of neurons per ganglion in the myenteric plexus under these colonization conditions (new Figure 1—figure supplement 2M). We thank the referees for this suggestion.

4) A better explanation of how curli exacerbates the motor phenotype would help. Is it through enhancing inflammation or affecting DA metabolism or affecting other areas that express high levels of α-synuclein in these transgenic mice? Is there a correlation between aggregate burden or inflammation and behavior?

This is indeed a critical point and we have addressed the various ways curli may impact motor symptoms in the manuscript (through modulation of inflammation and/or through acceleration of αSyn aggregation), though disentangling the pathways downstream of curli-mediated αSyn aggregation is not trivial. It is important to consider that αSyn aggregates activate the same innate immune receptors as CsgA (Kim et al., 2013; Tukel et al., 2005), and that inflammation can increase the propensity of αSyn to aggregate (reviewed in Lema Tome et al., 2013). Therefore, αSyn amyloid formation and inflammatory processes are intrinsically linked, and likely both involved in the disease process in ways that are currently unknown for any Syn-based animal model.

We have carefully revised the text in the discussion to unambiguously state that our results do not distinguish between inflammatory or amyloid-mediated outcomes. Further, we are actively addressing this topic by generating new mouse lines lacking specific inflammatory, αSyn aggregation or dopamine pathways to explore how curli exacerbate motor phenotypes.

[Editors' note: further revisions were suggested prior to acceptance, as described below.]

[…]However, there is a major point that must be addressed with necessary revisions to the text to tone down the relevance of their animal model and findings in relation to PD given the inability to reverse the phenotype with L-DOPA in this model (Point #1). An additional point is raised concerning the full immunoblot (Point #2). Please read below:1) The authors cited prior published work that the behavioral deficits in the Thy1-αSyn mice are not responsive to L-DOPA and as such argue that this experiment will not provide useful information. While we agree with the authors that in light of these prior published works, L-DOPA experiments are not likely to provide useful information. However, the fact that L-DOPA does not have an effect on the Thy1-αSyn mice indicates that the authors' claims that behavioral findings are due to exacerbation of PD-like motor phenotypes is specious. Since L-DOPA does not rescue the phenotype, while we understand their arguments, this animal model is not reflective of PD. Yet the authors continue to make statements throughout the revised paper that the motor deficits are consistent with the hallmarks of PD. For instance, the Abstract states "*Escherichia coli* exacerbates hallmark disease features such as motor impairment and αSyn pathology in the gut and brain." The authors need to therefore take a more balanced approach and present the Thy1-αSyn mice as a model of αSyn pathophysiology with accompanying behavior deficits and they should not make claims regarding motor impairment and the link to PD. In other words, the Thy1-αSyn mice represent a model of αSyn induced behavioral abnormalities only and claims regarding PD should be removed from the paper.

We appreciate the comments and agree that eliminating any extrapolation from the mouse model to Parkinson’s disease is appropriate, and have done so. However, it is important to consider that while L-DOPA responsiveness is a necessary component of PD, other pathological processes are occurring prior to- and concurrent with- loss of dopamine neurons (e.g., αSynaccumulation, mitochondrial dysfunction, impaired proteostasis). Thy1-αSynand other mouse models allow exploration into other synucleinopathy-associated pathologies.

Based on the editorial feedback, we have carefully revised the manuscript to present the mouse model in a more balanced fashion. Specifically, we have removed almost all mention of PD in the introduction and Results sections, and importantly no longer refer to the characteristics of Thy1-αSynmice as “PD-like”. Text previously focused on PD is now more broadly framed in the context of synucleinopathy and we discuss the mouse model as being relevant for understanding physiological impact of αSynaccumulation. The few instances where we retain mention of PD reflects the fact that there is simply much more research on PD and the gut microbiome than other related disorders. Accordingly, most if not all use of “PD” is confined to referencing published literature, mostly in the Discussion section, while being careful to not give the impression that we are modeling PD in our study.

2) We could not easily find the requested full immunoblot of soluble vs. insoluble α-synuclein fractionations. It's currently in an embedded tab of an excel file: is there a way to post this on the actual figure instead of as a source data file (Figure 1—figure supplement 2D)?

We have added text to the figure legend to state explicitly that full length immunoblots are available in the associated source data file, as it would take considerable effort to reformat an existing figure. If the manuscript is accepted, the source data files are easily accessed with a hyperlink for each figure in the formatted text.